# Territorial and Gender Differences in the Home Care of Family Members with Dementia

**Sagrario Anaut-Bravo and María Cristina Lopes-Dos-Santos ***

Department of Sociology and Social Work, Public University of Navarra, 31006 Pamplona, Spain;
sanaut@unavarra.es
* Correspondence: cristina.lopes@unavarra.es

**Abstract:** The increasing prevalence of dementia is threatening the capacity of health and social service systems to provide long-term care support at the territorial level. In both rural and urban areas, specific family members (gendered care) are responsible for the daily care of their relatives. The aim of this work is to explore gender and territorial implications in the provision of in-home care by family members. To this end, family caregivers in Navarre, Spain, were administered the Psychosocial Adjustment to Illness Scale (PAIS-SR) and a semi-structured interview. The results show the good psychosocial adjustment of caregivers of relatives with dementia but the negative impacts of caregiving in the domestic, relational, and psychological domains. Moreover, the feminization of psychological distress was found to predominate in rural areas since mainly women are responsible for instrumental and care tasks, while men seek other complementary forms of support. Place of residence (rural vs. urban) was found to exert a strong effect on the respondents' conception, life experience, and provision of care. Consequently, territorial and gender differences in coping with and adjusting to care require the design of contextualized actions adapted to caregivers' needs.

**Keywords:** family caregiver; dementia; psychosocial adjustment; rural–urban spaces; gender; aging; Navarre

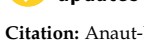



## 1. Introduction

Current evidence on family caregivers' adjustment to dementia, among other diseases [1–3], has shown that among them, place of residence and sex/gender are increasingly important variables to consider in this process.

In Spain, studies addressing psychological adjustment in the context of in-home care must be framed in an extensive tradition of both regional and urban studies linked to historical and political events that continue to resonate today [4–6]. Indeed, the country's autonomous government model, established in the Constitution of 1978, has produced significant social, economic, fiscal and cultural inequalities that warrant in-depth studies at the regional and municipal level, as well as comparative analyses of territorial differences in daily activities, such as caregiving for a relative. Since Spain joined the European Union, comparative analyses have been performed on both neighboring and non-EU states, including a growing number of publications that examine welfare systems and the welfare state model itself [7–9].

In contrast to the extensive literature on economic issues, territorial policies, or regional imbalances in Spain [6], regional studies on quality of life and welfare in the country remain scarce. This approach is recent and crosses disciplines such as anthropology, sociology, and the health sciences. As it occurs in other countries, there are few comparative studies on the care of the elderly and people with neurodegenerative diseases in rural and urban spaces [10–13]. Most studies in this line have focused on urban localities of different sizes [14] or on rural regions [15–17], while others have examined the diverse meanings of rurality and the heterogeneity of rural areas [18,19]. The social, economic, and political

changes driven by globalization have led to the emergence of new scenarios, such as the so-called new rurality in the 1990s, which is dissociated from agriculture, poverty, and notions of backwardness or passivity. Indeed, the effects of these different processes of change question the validity of a definition of rural and urban reduced solely to locality size, population density, and the dominant economic sector [4,13,20–22].

According to Dickins et al. [23] and Quesada-García and Valero-Flores [24], around 70% of elderly people live at home. Specifically, Lopes [25] calculated that 62% of people diagnosed with dementia in Navarre, Spain, remain living in their own homes or in those of relatives. This reality justifies the need for territorial studies that address urban–rural differences in care, despite the conceptual limitations mentioned above [4].

Regarding studies from a gender approach, there is a vast body of literature on the feminization of care for family members of different ages [26]. Indeed, gender-based research in this field has intensified due to what is called the "care crisis" [13,27]. This crisis has been exacerbated by an aging population that requires long-term community care with a certain level of specialization [28,29] and impacts both family solidarity and sustainability [30–32]. Moreover, the progressive social reorganization of work and caregiving has not prevented the vulnerability of caregivers [33] or the influence of personal and situational variables on the process of coping and adjusting to care [3].

Analyses from a gender approach have also brought to light the increasing role of men in caregiving (Granados and Jiménez [34], Del Río Lozano [35], Aguilar, Soronellas-Masdeu and Alonso-Rey [36], Rodríguez, Samper, Marín et al [14], Mosquera et al [37], Martín-Vidaña [38] and Zygouri [39]). In general, these studies evaluate the impact of caregiving on the quality of life of the caregivers, differentiating between men and women. The approach is diverse since the increasing prevalence of neurodegenerative diseases has placed the provision of long-term care and those who provide in-home care in a prominent spot in the social and health policy agendas of most countries [40].

As recognized in the case of China [41], there are very few regional studies [42–44] that examine caregivers from rural and urban areas in relation to sex and gender differences. Even fewer studies have examined the adjustment of those who care for family members with dementia, jointly considering both the rural–urban environment and gender. This is undoubtedly the main contribution of this article, although it will not always be possible to extrapolate our results to other regions or countries due to socio-cultural differences.

A previous study carried out in Navarre, Spain, compared adjustment to disease among in-home caregivers of relatives with Parkinson's disease and dementia [45]. The authors concluded that the specific illness of the cared person, place of residence, employment status, and income, in that order, were the most influential variables. In a subsequent analysis focusing only on dementia and the effect of place of residence on the psychosocial adjustment and quality of life of caregivers of relatives with dementia [46], differences were found between residing in urban and rural areas. These differences were due to specific territorial characteristics affecting the availability of human, economic, and technical resources, as well as cultural factors related to understanding and coping with life circumstances. Based on the results, the authors hypothesized that such differences might be explained by the person's ability to adapt, cope with, and accept the meaning of caregiving tasks, and that these skills could be the result of assumed gender roles.

Following the above, this article aims to examine the interaction between place of residence and gender roles in psychosocial adaptation processes among people who care for cohabiting relatives with dementia. To this end, we analyze Navarre, a region of Spain that stands out for its distinct historical background; wide geographic, economic, and sociocultural diversity; and its own tax system and a differentiated welfare model. Given the particular characteristics of Navarre, several region-specific [47–49] and comparative cross-regional analyses have been carried out on the region, such as the recent studies of Anaut-Bravo [50,51], Pérez and Martínez [52], and García and Caballeira [53], on health and social services systems.

## 2. Materials and Methods

The region of Navarre (Figure 1) has been chosen for the analysis due to its climatic and geographic diversity that conditions the population distribution as well as the predominant economic activities in the territory. Navarre has a total population of 661,023 (as of January 2021), which is mainly concentrated around its capital of Pamplona and the surrounding areas (just over 50%). It is also one of the five Spanish regions with the best social protection system development rates [54]. In addition, the accessibility to the information sources needed for this study makes Navarre particularly suitable for our aims.

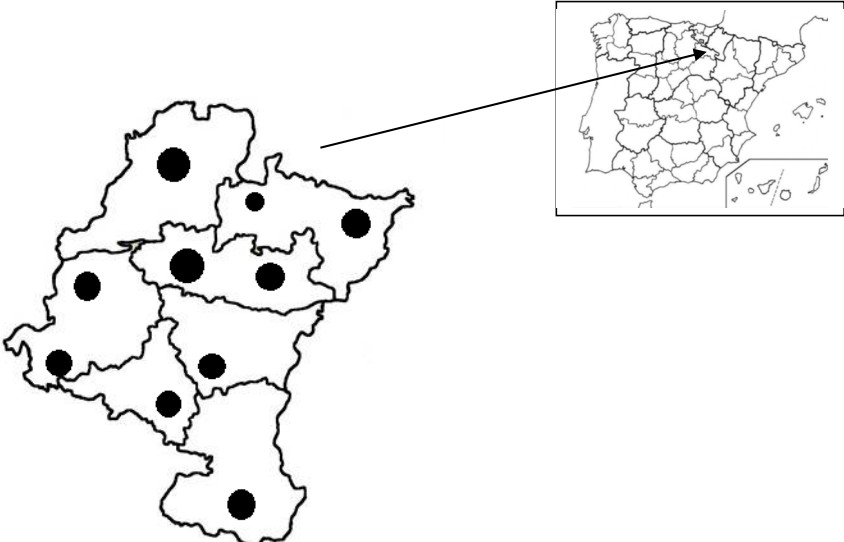

**Figure 1.** Territorial distribution of research participants.

The study sample was obtained (2015–2017) by non-probabilistic convenience sampling [55,56] since the research required the voluntary collaboration of primary health care and social service practitioners, as well as caregivers. A total of 61 practitioners involved in the Program for Dependency Care and Promotion of Autonomy of Basic Social Services and 55 practitioners from the Health Centers of Navarre agreed to cooperate in the research to identify and recruit family caregivers. All participants were provided the following documentation: basic information on the research content, an informed consent form, the two questionnaires that were administered for the study, the script of the interview, and the endorsement of the Ethics Committee of the Public University of Navarre (cod. PT-025-15). All of the practitioners were asked to confirm the diagnosis of dementia in the cared persons according to the International Statistical Classification of Diseases and Related Health Problems, 10th revision (ICD-10) [57].

A total of 135 telephone contacts were obtained, to which the following exclusion criteria were applied: being a relative without a direct care relationship, an occasional family caregiver, and limitations in verbal comprehension and expression. The inclusion criteria were being the main caregiver of the person with dementia and a cohabiting family member, good cognitive status to answer the questionnaire and take part in the individual interview, and voluntary participation and agreeing to sign the informed consent. The participants were from rural and urban areas and recruited at similar levels to their representativeness within the overall population of Navarre. Male and female caregivers were also recruited for the study (Table 1). The final sample was comprised of 75 caregivers of relatives with dementia residing in 29 localities of Navarre (Figure 1). Territorial representativeness was guaranteed in the distribution of participants.

In the sample obtained by the municipalities and population, participants from urban areas were clearly over-represented (Table 1), as were females. This was due to the voluntary nature of participation and some unforeseen circumstances, such as the hospitalization or illness of the family caregiver who was going to participate.

**Table 1.** Representation of municipalities and population of Navarre. * Rural: population, <10,000; ** urban: population, >10,000 [17]. Source: own elaboration based on data collected from the Statistics Institute of Navarre (NASTAT).

| Type of Locality | Number of Municipalities | Municipalities Percentage (%) | Population | Population Percentage (%) | Participants Percentage (%) |
|---|---|---|---|---|---|
| Rural * | 261 | 95.96 | 281427 | 43.75 | 18.2 |
| Urban ** | 11 | 4.04 | 361.807 | 56.25 | 81.8 |

Each of the 75 family caregivers were administered two questionnaires to collect quantitative information. The first one gathered the following sociodemographic characteristics of the participants: age, marital status, years caring for the family member affected by dementia, employment status, kinship, and level of education. The second questionnaire was the Psychosocial Adjustment to Illness Scale (PAIS-SR) developed by Derogatis [58] and validated by Bullinger et al. [59], which has been previously administered in Spain by Portillo et al. [60]. The PAIS-SR contains 46 items grouped into seven sections (Figure 2) to collect self-reported information on the evolution of dementia, changes in care as a result of the illness, alternatives to the mentioned changes, acceptance of and adjustment to dementia, factors of influence, and support networks and life satisfaction [61]. It should be noted that the results of Section IV of the PAIS-SR scale (sexual relations) were not analyzed due to the lack of responses (85%). However, the lack of responses was analyzed to ensure that the scale maintained its internal coherence according to Cronbach's analysis [62]. The data from the questionnaires were exploited using the SPSS Statistics program version 23 and R software R together with the integrated FactoMineR package [63] and Coheris SPAD.

To examine the personal assessment of care and process of adjustment to the disease in depth, a semi-structured interview script was prepared for this research following the seven sections of the PAIS-SR self-reported questionnaire (Figure 2). Sixty of the 75 caregivers of family members with dementia responded by reaching theoretical saturation and meeting the criterion of five interviews per analyzed variable (6 sociodemographic variables and 1 corresponding to each of the scales) of Peduzzi et al. [64].

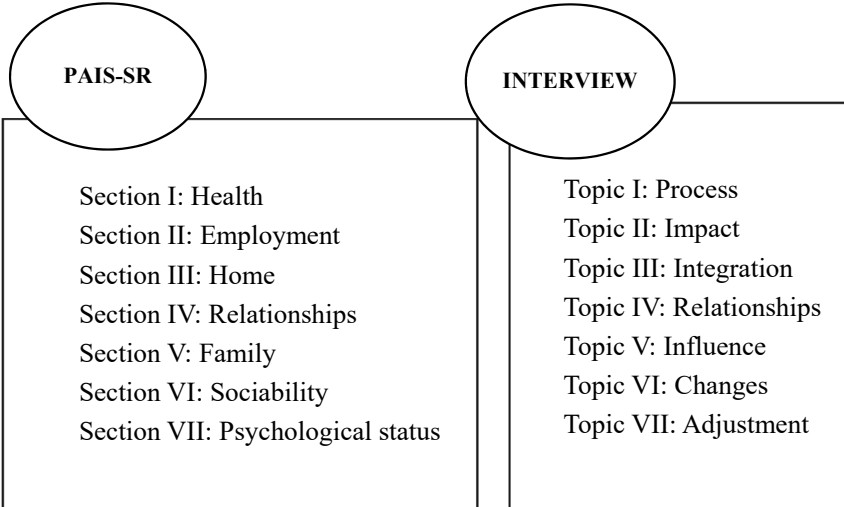

**Figure 2.** Equivalence of the contents of the PAIS-SR and the semi-structured interview. Source: own elaboration based on Portillo et al. [60].

The interviews were anonymized using the following codes: I (interview), F (family), M or F (male or female), R or U (rural or urban), and interview number. The data were analyzed using AQUAD 7 software. The transcripts (.docx) were transformed into .txt format files to extract the story structure and frequency of terms. The process proposed by Miles et al. was used to analyze the data [65].

## 3. Results

### 3.1. Family Caregiver Profiles

The general profile of the family caregivers is characterized by a historical trend: a greater number of females, married (64.7% men and 74% women), unemployed (70% men and 55% women), average age over 50 years, and basic level of education (51.6% women). However, differences by place of residence were detected.

In the rural areas, married caregivers predominated (Table 2), especially among women who were homemakers or engaged in full-time employment (same proportion). Additionally, male caregivers from both rural and urban area were, on average, older than female caregivers (up to 12 points of difference). It is interesting to note that the male caregivers were children of the cared person, not spouses. Regarding education, the men had only basic or vocational education, while the women had completed all educational levels, particularly basic education, followed at a certain distance by higher education.

**Table 2.** Sociodemographic variables of family caregivers; number by location. Source: own elaboration.

|  | Urban Localities | Rural Localities | Total |
|---|---|---|---|
| **Sex** |  |  |  |
| Male | 22 | 9 | 31 |
| Female | 63 | 41 | 104 |
| **Marital Status** |  |  |  |
| Married men | 15 | 6 | 21 |
| Married women | 43 | 31 | 74 |
| Single men | 5 | 3 | 8 |
| Single women | 11 | 9 | 20 |
| Men other | 2 | 0 | 2 |
| Women other | 9 | 1 | 10 |
| **Employment Situation** |  |  |  |
| Men full-time | 7 | 4 | 11 |
| Women full-time | 16 | 9 | 25 |
| Men part-time | 0 | 0 | 0 |
| Women part-time | 13 | 7 | 20 |
| Retired men | 13 | 3 | 16 |
| Retired women | 13 | 7 | 20 |
| Male homemakers | 0 | 0 | 0 |
| Female homemakers | 16 | 14 | 30 |
| Men other | 2 | 2 | 4 |
| Women other | 5 | 4 | 9 |
| **Education** |  |  |  |
| Men basic education | 7 | 4 | 11 |
| Women basic education | 29 | 25 | 54 |
| Men vocational education | 4 | 5 | 9 |
| Women vocational education | 7 | 4 | 11 |
| Men secondary education | 4 | 0 | 4 |
| Women secondary education | 13 | 4 | 17 |
| Men higher education | 7 | 0 | 7 |
| Women higher education | 14 | 8 | 22 |
| **Kinship** |  |  |  |
| Sons | 13 | 7 | 20 |
| Daughters | 48 | 24 | 72 |
| Male spouses | 9 | 2 | 11 |
| Female spouses | 11 | 13 | 24 |
| Men other | 0 | 0 | 0 |
| Women other | 4 | 4 | 8 |

Regarding participants from urban areas, most were married, especially the men. The majority of female and male participants had a basic level of education and an equal percentage of men had completed basic and higher education (23.5%). In terms of employment, most of the men were retired, while the women were engaged in full-time employment, followed at a short distance by those who were retired.

Regarding the socioeconomic differences between rural and urban areas, 47% of the male caregivers were retired and 88% lived in urban areas, while 15.6% of the female caregivers were retired and one in five lived in a rural location. Moreover, 45% of the women were employed, while 30% described themselves as "homemakers". No differences were found between urban and rural homemakers in terms of their representativeness but differences were detected in working time, with a higher percentage of full-time (56%) and part-time (70%) employed women in urban areas.

The higher employment rates of women (84.4%) point to the so-called "double shifts" and their higher qualifications. Of the female participants, 26% had a vocational or secondary education, while 22% had completed higher education, 64% of whom resided in urban areas.

### 3.2. Adjustment to the Care of Family Members with Dementia

#### 3.2.1. Care Experiences

Three categories related to the participant's care experiences were identified in the discourse analysis of the interviews: coping with coexistence, available resources, and harmonious coexistence (Figure 3). The three categories reflect the caregiving process and experience, external sources of support and the emotional impact, and the management of care. Both the men and the women who cared for relatives with dementia stated that coping with the situation is difficult ("hard") and has mainly negative impacts on their relational life in the form of obligations, poor health self-care, stress, lack of freedom, interdependence, and loneliness, among others. Negativity is also associated with unstable "life changes" and the "uncertainty" caused by the evolution of dementia when coping with coexistence situations.

However, there are some differences. For women in both rural and urban areas, continuous and exclusive dedication to care leads to greater personal and social isolation. By contrast, men give higher priority to their friendships and personal space.

IFFR11: "You don't feel like going out like you used to. When they come looking for you, you don't feel like it".

IFFU12: "Nobody ever comes, no one, no one".

IFMR43: "I need to go out with my friends; be with them".

IFMU61: "Keep my space, my moments, my places, my responsibilities and tasks".

In the case of women, there is an additional nuance regarding these gender differences: a certain traditional woman-to-woman solidarity seems to endure in rural areas. Such solidarity does not appear to be based on kinship relationships but on shared experiences and physical proximity. In contrast, feelings of loneliness and stress are more common among urban women, as stated by IFFU11: "Even if you are accompanied or there are people around, you feel a bit lonely".

Something similar occurs in family relationships, particularly intragenerational families. However, clear differences were not detected between caregivers residing in rural and urban locations or between men and women. The men tended to go straight to the point and in less detail, as IFMU028 succinctly stated "Family chaos," unlike women, who provide more details.

IFFR44: "Along the way, you get angry with your family because they don't want to deal with the problem. You also see yourself without a family!"

IFFR57: "I don't have a family to turn to. [It's] too much of a burden! We don't have a life together as a couple anymore".

IFFU19: "With my sister being ill, we had many differences. If there is good communication between family members, it's easier to deal with the disease".

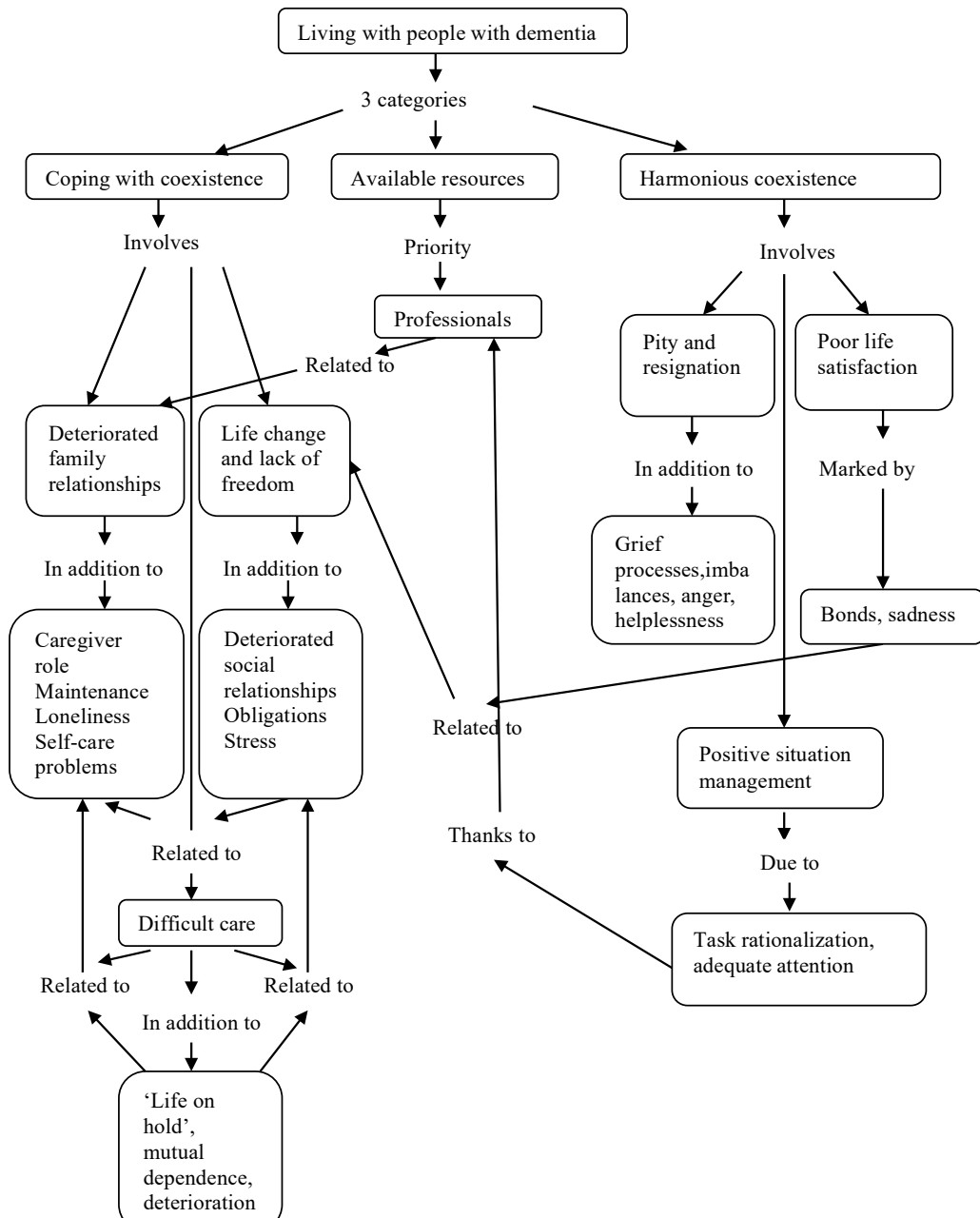

**Figure 3.** Conceptual map created from the discourse analysis of interviews with family caregivers. Source: own elaboration.

The discourse of family caregivers revealed the progressive deterioration of family relationships (Figure 2) and the importance of having both family support and complementary resources or services to cope with and adjust to dementia. Depending on the case, a combination of strategies may be used, such as family caregivers and support or external services (social services portfolio and/or formal caregivers). Given that there is always a family caregiver, professionals mostly provide any additional support and other family members are not involved in giving care, as shown in Table 3. This is consistent with the deteriorated family relationships mentioned by 65% of those interviewed.

**Table 3.** Source of care for family members with dementia; percentage distribution. Source: own elaboration based on data drawn from the interviews.

| Care Provenance | Percentage (%) |
| --- | --- |
| Professional | 50 |
| Cohabiting caregivers | 36.7 |
| Relatives | 13.3 |

Female caregivers from both rural and urban areas highlighted how the disease and the time they spend providing care has affected their relationship with their partners as well as produced the tremendous burden they must bear.

IFFR44: "Me, at home, my husband is very supportive, but you do get overwhelmed sometimes, the situation gets to be too much".

IFFU12: "We have very good communication with each other, but the couple suffers".

Men, however, tended to be more resolute, among other reasons, out of their desire to maintain their own space and activities, as mentioned above. Their views on the situation are similar to those of IFMU61: "I tend to seek solutions". The situation is resolved by means of external services that do both the domestic tasks performed by women prior to the illness and the specific care tasks. There are no differences by place of residence in regard to this question, nor in recognizing—as 52% of those interviewed stated—feelings that they have lost their freedom or that their life has been put on hold (Figure 2).

IFMR36: "At home things are as usual, not much has changed. What she didn't do is done by the girl who comes to clean and cook".

IFMU56: "The two hired people come, you pay them, and they help you. They are here in two shifts, and they do a lot".

Nonetheless, living in a place with available services introduces certain nuances not only in terms of the caretakers' attitudes, whose responses are conditioned by their gender, but also due to the possibility of obtaining this additional support. In this regard, women living in urban areas recognize the importance of municipal home care services (Servicio de Atención Domiciliaria, SAD) and adult daycare centers to better cope with the situation:

IFFU12: "The SAD comes two hours a day, five days a week. That help has been a lifesaver".

IFFU42: "He's in a daycare center. It's been a good thing".

Using the available public services or hiring outside help makes it easier for family caregivers to build, at least in part, their own life project, take care of tasks in accordance with their life stage, as well as ease the tension or stress and excessive work burden that caregiving involves (35% of respondents). As might be expected, the experience of those who live in rural areas is very different.

In rural environments, the scarcity of specialized social services is compounded by problems of accessibility in both their own and other localities. Mobility is often restricted by physical and geographical barriers or architectural barriers in housing, streets, and other public spaces. In some cases, the respondents also mentioned the lack of accessible public and private transport.

IFFR31: "We don't go out because we can't. We already had to take him out the window with a pulley once because he got sick".

The alternative in these cases is to hire outside help, but, as indicated by EFMR011, this only occurs when the family caregiver is no longer able to provide care: "I suffered from back pain for a long time; then we thought about hiring another person".

More than 58% of family caregivers stated that they are very or somewhat dissatisfied with their lives not only because of the impact of caregiving on their relational life but also on their emotional state (Figure 2). They find it increasingly difficult to live with their ill relative, which arouses mixed feelings of pity, resignation, helplessness, anger, rage, and anguish (47% of respondents).

IFFR11: "I felt very sorry for my mother and all I did was care for her".

IFFU12: "I get angry a lot; I get angry with myself; I get angry with her. There are days when you just want to escape from this life".

IFFU46: "It's maddening; It makes you angry, because everything revolves around the illness".

IFMR43: "Above all else, your character changes".

IFMU60: "A little helpless".

Caregivers who perceive they are coping with the situation in a coherent and adequate manner report greater life satisfaction with their tasks since they experience lower levels of stress and guilt, and demand less of themselves (Figure 2). However, 46.7% referred to how they managed the situation as "unstable" or "negative". In the interviews, no significant differences were found in this regard between men and women, or between those living in rural and urban areas. This seems to indicate, as the following comments suggest, that the key to coping lies in the skills, personality, and attitude of the family caregiver rather than in external factors.

IFFR44: "There are many days when I wake up and say: How well I am doing everything! Everything is going so well! Other times things don't work out so well".

IFFU50: "Sometimes [it's] good, sometimes [it's] crazy".

IFMR43: "I've taken everything that was bad in the house to turn it into something good, get rid of the stress and increase my energy, and do something. I'm satisfied. Some (things) are good, and others are bad, but if you have a balance of both and nothing stands out more, you swim in a calm sea, since you're okay".

IFMU60: "It's more like she's in command and you're just running behind her trying to figure out what's going on".

It is important to note that more than half of the family caregivers positively valued (or at least not in an overtly negative way) how they handled living with a family member with dementia. These caregivers did not usually suffer from general or permanent psychological distress. Opinions such as those indicated below demonstrate that the caregivers view dementia as an irrevocable fact to be accepted and a reality that will be part of their daily life for some time. They do not feel resignation but rather accept a reality. Acceptance is expressed in comments such as that of EFVR043: "I am satisfied, I wouldn't ask for more or for less".

IFFR07: "You could see it coming. With age and time these things happen".

IFFU09: "It's a question of accepting that you'll need help and that it'll get worse over time".

The three categories analyzed (Figure 2) indicate that female caregivers suffer episodes of psychosocial distress as a consequence of their greater dedication, while male caregivers employ strategies to share and delegate tasks to external help. No notable differences were detected in terms of assumed gender roles between urban and rural participants. These territorial differences were reflected in options for accessing resources to support caregiving tasks either in the form of public services or external personnel.

### 3.2.2. Coping with and Adjusting to Caring for a Family Member with Dementia

The results of the PAIS-SR corroborated the findings of the interviews. The Holm's test ($p < 0.001$) indicated that the three most significant sections of the PAIS-SR are the domestic environment (Section III), the social environment (Section VI) and psychological distress (Section VII). As Table 4 shows, urban caregivers have a negative perception of all three environments, while rural caregivers only perceive the domestic and social environment negatively. Regarding psychological distress (Section VII), in particular, urban caregivers manifest higher levels than rural caregivers, thus suggesting that rural caregivers adjust better to caregiving.

**Table 4.** Mean scores for Sections III, VI and VII of the PAIS-SR by place of residence. * The maximum score on the PAIS-SR is 24 points for Section III, 18 points for Section VI and 21 points for Section VII. Lower scores indicate better psychosocial adjustment, and higher scores indicate poorer psychosocial adjustment. Source: own elaboration.

| PAIS-SR * | Urban | Rural |
|---|---|---|
| Section III: Domestic environment <12 good perception | 12.6 | 12 |
| Section VI: Social environment <9 good perception | 10.2 | 10.6 |
| Section VII: Psychological distress <10.5 no presence of discomfort | 10.1 | 4.4 |

The mean scores presented in Table 4 were further disaggregated by both sex and place of residence to obtain more specific data regarding the impact of caregiving (Table 5). In general terms, men from rural areas have a better perception of the domestic and social environments and suffer less psychosocial distress. Compared to women, men residing in urban and rural locations adjust better to the domestic environment. Furthermore, men and women residing in urban areas show very similar scores overall, except in the social environment, which women view more positively. The results are similar for psychological distress among those living in rural areas. Therefore, those who live in rural areas generally have a more positive perception and, within this group, men cope with and adjust better to caregiving than women.

**Table 5.** Mean scores for Sections III, VI and VII of the PAIS-SR by sex and place of residence. Source: own elaboration.

| $\bar{x}$ | Domestic Environment Section III | Social Environment Section VI | Psychological Distress Section VII |
|---|---|---|---|
| **Men** | | | |
| Rural | 11.8 | 9 | 4.55 |
| Urban | 12.12 | 11.12 | 10.15 |
| **Women** | | | |
| Rural | 12.47 | 10.81 | 4.76 |
| Urban | 12.41 | 9.36 | 10.32 |

Regarding the variances in the PAIS-SR, the most marked differences were found in all sections for caregivers residing in urban areas, indicating that the variability in caring for a family member with dementia is more pronounced in urban areas (Table 6). By sex, significant differences were observed in the domestic and social environments. In both these environments (Sections III and VI), women living in urban areas showed the greatest discrepancies with respect to the mean. The same discrepancy was found in Section VII (psychological distress), although with less intensity. Women in rural areas and men in urban areas also showed this variability in responses but to a lesser extent. Men residing in rural areas showed an almost homogeneous perception in their responses, although in psychological distress, those residing in urban areas also showed a very small deviation.

**Table 6.** Variance in PAIS-SR Sections III, VI and VII by place of residence and sex. Source: own elaboration.

| $S^2_x$ | Domesticenvironment Section III | Socialenvironment Section VI | Psychologicaldistress Section VII |
|---|---|---|---|
| **Men** | | | |
| Rural | 3.33 | 0 | 0.25 |
| Urban | 7.55 | 6.12 | 0.98 |
| **Women** | | | |
| Rural | 9.04 | 5.92 | 2.81 |
| Urban | 13.22 | 10.76 | 10.03 |

Therefore, the variables territory and sex captured explicit differences in how the respondents interpret and understand the care of a family member with dementia. Gender roles and interaction with the environment (services, resources, relationships, etc.) are explanatory factors for such differences, as determined by the qualitative analysis of the discourse.

## 4. Discussion

The results reveal that the processes of coping and adjusting to dementia among family caregivers in Navarre differ according to place of residence (rural or urban) and sex. However, when considering all respondents, the findings indicate relatively good psychosocial adjustment (PAIS-SR), especially among caregivers living in rural areas. This result is consistent with Ehrlich et al. [1], although the authors found that satisfaction with caregiving occurs in urban and not rural localities, as has been [10] previously demonstrated.

Various studies have shown negative psychosocial adjustment in cases of long-term care and dementia [66–68]. The vast medical and nursing literature has also highlighted these negative impacts [69,70], especially on the health of family caregivers [35,71,72]. Other studies have focused on negative social and family impacts, such as stress, sleep problems, or the loss of personal independence, future expectations, and social relationships [3,73–75]. The respondents in our study referred to impacts of a social nature but attached less importance to their own health issues.

Despite the generally positive process of psychosocial adjustment among the respondents, the PAIS-SR and interviews indicated three domains in which the provision of care has a negative impact on family caregivers: the domestic, the relational, and, to a lesser extent, the psychological domains. Regarding the domestic domain, no significant differences were found between urban and rural locations. An explanation for this could be the progressive social, cultural, and economic rapprochement due, among other reasons, to greater territorial mobility [76]. This result may also be explained by the increasingly disperse family networks, weak social networks, smaller families, and the feminization of urban locations compared to the masculinization of rural ones. All these factors have led to the defamiliarization of care and limited the capacity of families to care for elderly patients who want to stay at home, as Prieto [19] and Martín and Rivera [17] have argued.

Given such changes, it is important to analyze coping strategies and adjustment in caregivers of family members with dementia from the viewpoint of psychological distress/well-being. While the PAIS-SR responses showed significant differences by place of residence and sex, in the interviews, the respondents placed greater emphasis on psychological distress, which was more pronounced depending on the caregiver's gender. Specifically, more women stated that they experienced psychosocial distress, particularly those living in urban settings. According to Losada et al. [77], the feminization of distress is due to women's greater emotional involvement and heavier care burden, which leads to a state of hypervigilance. In turn, hypervigilance affects caregivers' emotional attachment to the cared family member, as well as their life satisfaction and health status, as previous studies have shown [35,37,39,72]. The different types of bonds between the ill person and

family members (i.e., spouses, daughters, and sons) detected in this study are in line with Rodríguez and Pérez [3].

An important factor in bonding is the disease itself [45]. Considering dementia involves the dynamic and progressive deterioration of cognitive function, it requires a significant effort of adjustment, flexibility, and resilience on the part of caregivers [2,3,69]. As gleaned from the interviews, caregivers' perception of being overburdened converges with the course of the disease and varying degrees of psychosocial distress, as well as the belief that professional and non-professional support for in-home care is not and will never be sufficient.

Regarding gender differences in psychosocial adjustment, our results diverge from those of Rodríguez et al. [14] as we found that certain aspects of the coping and adjustment strategies deployed by men differ from those of women. Firstly, while women assume instrumental and care tasks in their entirety, men seek complementary support more quickly. In other words, men adopt more flexible coping and adjustment strategies that reduce their psychosocial distress, as also demonstrated by Zygouri et al. [39]. Secondly, men have a more pragmatic attitude and self-manage their time and activities to not renounce their social relationships. Thirdly, care involvement by men in rural areas may be related to three features that characterize the smallest localities in our study and which coincide with other regions of Spain such as Castile and León [17]: the masculinization of the population, men's singleness, and the increased life expectancy of males. For this reason, many men living in rural areas are responsible for caring for their parents. However, when possible, they opt for external support to continue working (those who are retired continue to work their land) and enjoy a certain social life, knowing that support from family members is scarce and sporadic. The involvement of males in the care of ill family members has been detected for some decades and seems to be related to age [78]. For Spain, Abellán et al. found that there are more male than female caregivers in all types of households and forms of care from the age of 80 [79], which may explain the high average age of male caregivers in the rural areas of Navarre (64.6 years old).

The relationship between psychosocial distress and gender cannot be explained solely by weak social networks and assigned gender roles [80], nor by the fact that family support is more common in rural areas, as Manso et al. [81], Lorenzo et al. [82], and Ehrlich et al. [1] have argued. The environment where one lives and the available opportunities for biopsychosocial adjustment must also be considered [83].

Until recently, place of residence was not believed to play a significant role in caregivers' experiences. Yet, the interviews we conducted clearly indicate that the availability of services to aid in the care of ill family members, as well as family and non-family support, depend on where one lives [46]. Likewise, the physical location and geography of the place of residence affects the municipality's own accessibility (public spaces) as compared to other municipalities that offer more services [84]. In rural locations, this may partially explain the limited social and family relationships identified in the PAIS-SR, but not the lack of such support mentioned by the respondents in the interviews. In this case, both urban and rural family caregivers coincide in their assessments.

The lack of social and family support has also been reported in other studies, such as Martínez for Spain [32], Rubio et al. for Chile [75], and Wang et al. for China [41], who examined the decline in family solidarity and changes in social relationships that weakened support networks. In the interviews, the respondents highlighted problems such as family conflicts, the lack of support from other family members, and the negative impact of caregiving on relationships with other members of the nuclear family. This focus on the family rather than on social networks is related to what is called the "Mediterranean care model", that is, high family involvement supported by little formal care [2].

However, weak family, social, and community support does not seem to affect emotional and cultural ties to the geographical place in which one has lived and wants to live. There is a personal connection with the environment, that is, people build their daily lives together with others who live in the same social and territorial context to create a culture

and an identity. Nonetheless, this permanence of place is only possible if access to certain professional services is available.

Around 50% of the cases studied have complementary support resources in the form of either public services (SAD or adult daycare centers) or financial aid to hire non-family caregivers through the social services portfolio of Navarre. The recognition of dependent benefits under the Dependency Law (2006) may be key to increasing such support in both urban and rural areas. However, the most important problem mentioned by both urban and rural respondents is the lack of services. This is striking given the greater number of specialized dementia care services, options for hiring non-family caregivers, and resources to support caregivers in urban areas, as Martín and Rivera have shown [17]. In addition to the scarcity or lack of services in rural municipalities, family caregivers face the problem of accessibility to neighboring municipalities where such services are available.

None of the respondents questioned the decision to remain at home as it was an essential part of their lives. Living in a familiar environment (one's home, neighborhood, or municipality) provides both caregivers and the cared person emotional and relational support, as well as a sense of personal identity. Nonetheless, more accessibility, flexibility, services, social networks, etc., are needed to adapt to the increasing presence of elderly people with neurodegenerative diseases, particularly in rural areas. These results are in line with the characteristics of rural environments pointed out by Prieto [19]: aging and over-aging; scarce and difficult access to services, infrastructures, and ICTs (Information and Knowledge Tecnologies); disperse family networks (due to emigration); the defamiliarization of care (hiring of non-family caregivers); and the masculinization of the family caregiver figure. In this sense, the territorial and social context is key to understanding each person's own coping process.

The present study has two main limitations. The first is the need to increase the size and geographical scope of the sample. In the final sample, urban dwellers and females were over-represented, which may have biased the quantitative results. The second limitation is the need to examine in greater depth the factors that most impact on care-related issues, such as gender, marital status, and the specific characteristics of households and public spaces. These and other factors play an essential role in caregivers' psychosocial adjustment to the disease and the development of collective coping strategies to ensure the sustainability of care. Despite these limitations, this article offers an adequate framework for conducting comparative studies between and within regions in the future.

## 5. Conclusions

Processes of psychosocial adjustment among family caregivers differ depending on their place of residence and gender. While women tend to assume the responsibility for long-term care and their socio-emotional involvement hinders social interactions and increases their emotional burden, men cope with the situation in a more pragmatic manner to maintain, to the largest possible extent, their habits and social relationships, as do women who work part or full time. Indeed, working outside the home improves the coexistence with the ill family member and reduces both the psychosocial distress and stress of providing care, despite negative feelings associated with the belief that they are not providing sufficient care.

This study has also explored the territorial dimension of psychosocial adjustment in the care of family members with dementia. The desire to remain at home speaks to notions of rootedness, identity, security, and certainty regarding private (home) and community (local) spaces. Such life choices can have impact on rural environments since they are characterized by demographic aging (the majority of inhabitants are over 65 years old) and depopulation, and many are at risk of disappearing.

However, rural environments are also defined by vital attitudes, such as the acceptance of the normal course of life and nature that helps to reduce psychosocial distress in caregivers of relatives with dementia, and greater resilience, as the interviews have shown. Moreover, not only do those who need care remain but also those who take care

of them, whether they are relatives or not. In this way, the cared person and the caregiver contribute to the continuity of the local population and slow down depopulation. The results presented in this work also show that caring for a family member with dementia in rural areas is associated with lower levels of psychosocial distress than in urban locations, where caregivers often experience feelings of social isolation despite the availability of complementary care services. In fact, caregivers in rural areas—especially men—experience less psychosocial distress despite the fewer professional social and health services and the decline in family support networks, as they are compensated for by other types of resources.

Although the results point to gender differences in how in-home family care is conceived, experienced, and provided, new trends are emerging in rural localities. In this regard, neither age nor being a man appears to be a restriction for the provision of care. Indeed, the so-called "new caregivers" (men over 80 years old) demonstrate that caregiving is no longer synonymous with the loss of biopsychosocial health or poor quality of life but is more closely associated with personal and family coping strategies and adjustment to the care of family members with dementia.

Based on the above considerations and given the increasing prevalence of neurodegenerative diseases such as dementia, it will be necessary to design contextualized actions aimed at meeting caregivers' needs. These actions should promote the comprehensive care of caregivers to ensure the sustainability of family care. To achieve this, multidisciplinary, adaptive, and community services from a social co-responsibility approach are required. Additionally, we must not overlook the impact of the new rurality on the rural environment in terms of caregiver profiles and behaviors, and the demand for services, among other aspects, which are comparable to those of urban locations. In other words, we find ourselves, as A. Moreno [77] states, before new non-dichotomous urban–rural relationships that are transforming social and cultural constructs.

**Author Contributions:** All authors should be involved. Conceptualization, S.A.-B.; methodology, M.C.L.-D.-S. and S.A.-B.; software, M.C.L.-D.-S.; validation, M.C.L.-D.-S.; formal analysis, M.C.L.-D.-S.; investigation, M.C.L.-D.-S.; resources, M.C.L.-D.-S.; data curation, S.A.-B. and M.C.L.-D.-S.; writing—original draft preparation, M.C.L.-D.-S.; writing—review and editing, S.A.-B.; supervision, S.A.-B.; project administration, S.A.-B.; funding acquisition, S.A.-B. All authors have read and agreed to the published version of the manuscript.

**Funding:** This research study was funded by Fundación Bancaria la Caixa y Fundación Caja Navarra, REF P/1/19.

**Informed Consent Statement:** Not applicable.

**Data Availability Statement:** The data that support the findings of this study are available from the corresponding author upon reasonable request. NASTAT is an open access website at: http://www.navarra.es/home_es/Gobierno+de+Navarra/Organigrama/Los+departamentos/Economia+y+Hacienda/Organigrama/Estructura+Organica/Instituto+Estadistica (accessed on 14 June 2021).

**Conflicts of Interest:** The authors declare no conflict of interest.

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
