# Peer review of "Territorial and Gender Differences in the Home Care of Family Members with Dementia"

_land, doi:10.3390/land11010113_

Round 1
Reviewer 1 Report
This is a work of great interest and utility. The text is written clearly, so it is easy to read. Obviously, I am referring to its quality in the way it presents its reasoning, not to its linguistic quality, since I am not a native English speaker and I not able to comment on it. In addition, the methodology used is adequate and the references cited are timely. However, some additional references are missing. The suggestion is that these new references are adequate to understand and explain the following issues that are considered in the manuscript:
First, more or less on line 216, it says “Both men and women who care for their relatives with dementia state that coping with such situation is difficult,“ hard ”and with fundamentally negative impacts on their relational life: obligations, poor health self-care, stress, lack of freedom, interdependence, loneliness, etc. Negativity associated with a "change of life" that does not stabilize, but moves within the characteristic "uncertainty" of dementias evoluton when managing coexistence situations. However, there are some differences. In the case of women, both in rural and urban areas, continuous and exclusive dedication to care leads to greater personal and social isolation. Men, for their part, give higher priority to their friendships and their personal space. "
In this regard, the authors should examine and consider some of the many published works (articles or books) on the sociology of gender that are useful in explaining these differences. Differences that, to a great extent, are motivated by forms of education and socialization that produce and reproduce inequalities between men and women. When I say this, I am thinking of authors like María de los Angeles Duran, Bourdieu (When he talks about “male domination”), Constanza Tobío, Olga Salido, etc., etc. In my view, the reading and consideration of the aforementioned authors, as well as other authors like that, will be very useful to the authors of this manuscript in order to better understand the reasons why, unlike men, women, even when working outside the home (thus entering the public sphere), have fewer opportunities to avoid their role as caregivers in the home, as this role has been strongly internalized by the majority of society and of women as an obligation inherent to their feminine gender. As a consequence, they tend to be very self-demanding and to blame themselves when they feel that they have failed in their primary responsibility in cases where they deem that their home is not running well.
Secondly, given that the article talks about rural-urban territorial differences, the authors should take into account some additional works suitable for the delimitation of rural and urban areas, in addition to the document published in 2016 by the BBVA Foundation that they already cite from Reig, E .; Goerlich, F.J .; Cantarino, I. This document is available online: https://www.fbbva.es/wp-content/uploads/2017/05/dat/DE_2016_IVIE_delimitacion_areas_rurales.pdf., Specifically, authors of this manuscript should take into account other references with a more sociological orientation and, in my opinion, more appropriate to see what is rural and what is urban, the increasingly blurred physical and social borders between rural and urban, as well as the changing meanings of the rural and the urban temporally, socially and spatially. For example, authors should consider works by authors such as Camarero-Rioja, Entrena-Durán, Baigorri, Oliva-Serrano, Muñoz-Sánchez, etc., etc.
In addition to what has been said so far, there is a paragraph that is not clear what the authors mean. This is the paragraph that begins on line 43: “Globalization and the extension of a new rurality have introduced changes that question a stated definition reduced to the locality size, population density and primary sector economic predominance [13, 20]”.
I consider that this expression is inappropriate, basically for three reasons: a) because the authors do not explain in the manuscript what globalization is; b) what is new rurality is not explained either; c) It is not clear how what is stated in this sentence about globalization and the new rurality contributes to a better understanding of the rural / urban differences with regard to male and female behaviors before the care of family members with dementia at home.
Moreover, the authors say that they did 135 semi-structured telephone interviews. In this regard, it would be very convenient for them to add a new table to the manuscript, including a classification of the interviews carried out. Obviously, it is not feasible to include in that table a list of the 135 interviews carried out. In my opinion, Figure 1. On “Territorial distribution of research participants” is not enough to give the reader an idea of ​​how these interviews have been distributed throughout the territory under study. For this reason, the new table that I am suggesting to the authors should, among other things, group these interviews by profiles, classify them according to the urban or rural locations where they have been conducted and the sex of the interviewees.
Finally, it is very convenient that the authors make a meticulous and careful reading of the manuscript to detect and correct possible errors, such as the following two errors that I have found:
- Reference 13. Agulló-Tomás, Mª S .; Zorrilla-Muñoz, V .; Gómez-García, Mª V. Socio-spatial approach to aging and programs for caregivers of the elderly. International Journal of Developmental and Educational Psychology (INFAD), 2019, 1, 211-228. DOI: 10.17060 / ijodaep.2019.n1.v2.1433.
In this reference, instead of “Aproximación socio-espacial añ envejecimiento” it should say “Aproximación socio-espacial al envejecimiento”.
- On line 221 it says “dementias evoluton”, but it should say “dementias evolution”.
I hope my previous comments are useful and help the authors to improve their manuscript. I encourage them to go ahead and wish them all the luck, since, I reiterate once again, the manuscript is worth it, it is well done methodologically and, in general, well founded bibliographically, at the same time that it is undoubtedly useful for society and those responsible for implementing health policies, given the issue it deals with and the way it addresses it.
Author Response
Thank you very much for all your comments, suggestions and indications. We have carefully taken them into consideration to improve manuscript quality and clarity.
Response to reviewer 1 comments
1.1. The authors should examine and consider some of the many published works on the sociology of gender that are useful in explaining these differences.
The thinking on the provided sociology of gender is very appropriate. Within Results section, there are questions such as those that point to inequalities in work activity, educational level, roles, etc. Within Discussion section, direct references are made from different approaches to the questions posed by the proofreader which are included in the research results. We have understood that the focus of interest is not a detailed analysis from gender sociology.
In the Discussion, the following aspects are related: personal coping with the care of a relative, sex and/or gender and the residing territory. Land is more specialized in the territorial dimension and, therefore, we have sought a balance between the three categories of analysis. It is possible that this limits the level of deepening, but it is clear in the article that the territory in its physical dimension as well as its social and cultural dimension, affects and is affected by the social construction of gender. Thus, the dwelling place and the assumed roles have sufficient identity to establish differences in coping with the situation of caring for a family member with dementia.
A reference to Mª de los Ángeles Durán (reference 80) has been incorporated and that of Constanza Tobío (line 70, reference 43) was already included. There are also other publications, perhaps less well known, on the sociology of gender such as that of P. Mayobre and I. Vázquez (line 57, reference 28).
1.2. The authors of this manuscript should take into account other references with a more sociological orientation and more appropriate to see what is rural and what is urban. The authors should consider works by authors such as Camarero-Rioja, Entrena-Durán, Baigorri, Oliva-Serrano, Muñoz-Sánchez, etc., etc.
A more careful reading has been made regarding the recent publications of articles by the cited authors. They are all of them very interesting, they have been moved away from the arguments presented in the article. It is true that they allude to problems in the care of those who live in rural areas, but it would force the authors to carry out another type of analysis, which this time has not been performed.
However, two more references have been included that complete, we believe, the panorama:
- Camarero-Rioja, Cruz, Oliva, J., 2016 (reference 76).
- Goerlich, Reig, Cantarino, 2016 (reference 84).
1.3. There is a paragraph that is not clear what the authors mean. This is the paragraph that begins on line 43.
In order to clarify concepts (from line 61) information has been expanded, without entering into debates, because it is the Introduction.
1.4. It would be very convenient for them to add a new table to the manuscript, including a classification of the interviews carried out. The new table should group these interviews by profiles, classify them according to the urban or rural locations and the sex of the interviewees.
Such Table has been included following the recommendations.
1.5. It is very convenient that the authors make a meticulous and careful reading of the manuscript to detect and correct possible errors.
We have counted on the collaborative of a native English translator to review the whole text.
1.6. About errors (reference 13)
The publications in the journal INFAD (International Journal of Developmental and Educational Psychology) collected in References are in Spanish (example: nº 13), although the journal title is in English.
Reviewer 2 Report
The article is scientifically sound, and the research deep enough. The manuscript is of potential interest to the broad international audience of "Land". However, there are two major shortcomings, one related to the fact that the presentation does not seem to match the journal, and the second related to the poor level of English, incompatible with an international journal.
The first and most important relates to the general topic of the article, which seems to be misfit to "Land". In order to address this issue, the authors should elaborate more on the interpretation of their results from a territorial perspective, connecting them to environmental interpretations.
The second one relates to the very poor level of English, incompatible with an international journal and aggravated by the careless editing of the manuscript. Problems include awkward phrasing: "it predominates, especially in urban spaces, a feminization", instead of the normal one ("a feminization... predominates") (lines 15-16), "After the carried out analysis, it should be recognized two types of limitations" instead of "two types of limitations are recognized in the analysis carried out" (line 523), sentences lacking a verb: "Comparisons that, since Spain’s incorporation into the European Union, have been generalized..." (lines 31-32), lack of a period between two sentences in the same paragraph (line 35), and presence of text in other language than English (lines 268, 611). The authors should seek for the assistance of a native English speaker or even professional editing services. Please note that the flagrant examples above are just a selection; such issues are present all over the manuscript.
Author Response
Thank you very much for all your comments, suggestions and indications. We have carefully taken them into consideration to improve manuscript quality and clarity.
Response to reviewer 2 comments
2.1. The authors should elaborate more on the interpretation of their results from a territorial perspective.
Within the Discussion, the following aspects are related: personal coping with the care of a relative, sex and/or gender and the residing. We are aware that Land is a journal specialized in the territorial dimension and, therefore, we have sought a balance between the three categories of analysis. It is possible that this exercise limits the level of deepening, but we hope through the corrections made to clarify in the article that the territory in its physical as well as its social and cultural dimension, affects and is affected by the social construction of gender. Thus, the dwelling place and the assumed roles have sufficient identity to establish differences in coping with the situation of caring for a family member with dementia.
Sentences and paragraphs have been included in the Discussion and Conclusions sections as well as some clarifications on the Introduction, which emphasize the territorial dimension of results interepretation
2.2. The authors should seek for the assistance of a native English speaker or even professional editing services.
The authors made a meticulous and careful manuscript reading to detect and correct possible errors. A native English translator has already collaborated on such task.
Round 2
Reviewer 2 Report
The authors have addressed all my previous comments and, as a result, the manuscript gained research depth and addresses a broader international audience. I do not have any further suggestions and recommend the publication of the article in its revised form.